# Clinical Parameters Affecting the Therapeutic Efficacy of SGLT-2—Comparative Effectiveness and Safety of Dapagliflozin and Empagliflozin in Patients with Type 2 Diabetes

**DOI:** 10.3390/healthcare10071153

**Published:** 2022-06-21

**Authors:** Irina Claudia Anton, Liliana Mititelu-Tartau, Eliza Gratiela Popa, Mihaela Poroch, Vladimir Poroch, Delia Reurean Pintilei, Gina Eosefina Botnariu

**Affiliations:** 1Department of Pharmacology, Clinical Pharmacology and Algesiology, Faculty of Medicine, ‘Grigore T. Popa’ University of Medicine and Pharmacy, Universitatii St. 16, 700115 Iasi, Romania; irinaanton@taktfest.ro; 2Department of Pharmaceutical Technology, Faculty of Pharmacy, ‘Grigore T. Popa’ University of Medicine and Pharmacy, Universitatii St. 16, 700115 Iasi, Romania; 3Department of Family Medicine, Preventive Medicine and Interdisciplinarity, Faculty of Medicine, ‘Grigore T. Popa’ University of Medicine and Pharmacy, Universitatii St. 16, 700115 Iasi, Romania; boanca.mihaela@umfiasi.ro; 42nd Department of Internal Medicine, Faculty of Medicine, ‘Grigore T. Popa’ University of Medicine and Pharmacy, Universitatii St. 16, 700115 Iasi, Romania; vladimir.poroch@umfiasi.ro; 5Department of Diabetes, Nutrition and Metabolic Disease, Consultmed Medical Center, Pacurari St. 70, 700544 Iasi, Romania; drdeliapintilei@gmail.com; 6Department of Diabetes, Nutrition and Metabolic Disease, ‘Grigore T. Popa’ University of Medicine and Pharmacy, Universitatii St. 16, 700115 Iasi, Romania; ginabotnariu66@gmail.com

**Keywords:** type 2 diabetes, dapagliflozin, empagliflozin, antidiabetics, patients

## Abstract

(1) Background. We aimed to assess long-term efficacy and safety in inadequately controlled type 2 diabetes (T2DM) of two SGLT-2 inhibitors: empagliflozin (Empa) and dapagliflozin (Dapa), combined with metformin, other oral antidiabetics or insulin, according to the protocols in Romania. (2) Methods. The data of 100 patients treated for T2DM with associated dyslipidemia and/or cardiovascular diseases at the University Hospital and Consultmed Medical Center in Iasi were retrospectively reviewed (2017–2021). In total, 48 patients had received dapagliflozin (10 mg with oral antidiabetics or insulin) and 52 patients received empagliflozin (10 mg /25 mg with oral antidiabetics). (3) Results. In both groups, the lowering of BMI was significant: Dapa group (32.04 ± 4.49 vs. 31.40 ± 4.18 kg/m^2^; *p* = 0.006), and Empa group (34.16 ± 5.08 vs. 33.17 ± 4.99 kg/m^2^; *p* = 0.002). Blood sugar average levels decreased significantly (170 vs. 136 mg/dL; *p* = 0.001 for Dapa; 163 vs. 140 mg/dL; *p* = 0.002 for Empa) and also average levels of HbA1c (7.90% vs. 7.51%; *p* = 0,01 for Dapa; 7.72% vs. 7.35%; *p* = 0.004 for Empa). (4) Conclusions. Better results in all variables were observed in younger male patients with a shorter duration of diabetes and threshold BMI levels of 34.1, treated with SGLT2, and more significantly with Empa.

## 1. Introduction

Type 2 diabetes mellitus (T2DM) is a major chronic disease, a leading cause of morbidity and mortality with an alarming increase in prevalence throughout the world [1,2,3]. T2DM is characterized by a progressive resistance to insulin and hyperglycemia, as well as pathologic activation of the immune system, leading to a gradual impairment of microvascular and macrovascular tissue function [4]. For these reasons, treatment of T2DM can be pivotal to improving and maintaining a healthier lifestyle for affected patients. However, choosing an effective treatment regimen remains challenging. Currently, the available drug treatments cannot provide consistent control of the circulating glucose levels; thus, depending on the patient, the effect of slowing the onset and progression of cardiovascular and renal injuries is little [5]. However, the type and number of effective oral or injectable drugs that can be used to treat T2DM remain to be fully described, taking into account the broad spectrum of patients and their specific clinical needs [6,7,8,9].

Sodium-glucose cotransporter-2 (SGLT-2) inhibitors are a novel class of oral hypoglycemic agents for the management of type 2 diabetes mellitus (T2DM) [10]. In the past few years, there has been an interest in the development of SGLT-2i to improve glycemic control and manage weight reduction without increasing the risk of hypoglycemia [11,12,13]. In the class of SGLT2i, there are differences in oral bioavailability, plasma protein binding, metabolism, clearance and selectivity for the SGLT2 protein (canagliflozin being the least selective, while other derivatives such as dapagliflozin, ertugliflozin and empagliflozin are highly selective). Dapagliflozin and empagliflozin exhibit quite similar oral bioavailability (78% respectively 60%), but empagliflozin is twice as selective as dapagliflozin (2500-fold vs. 1200-fold compared to SGLT1i action) [14]. SGLT-2 inhibitors decrease the fasting plasma glucose (FPG) postprandial plasma glucose levels, and the level of glycosylated hemoglobin (HbA1c), in an insulin-independent manner, in addition to reducing body weight via secondary caloric loss due to urinary glucose excretion [15,16,17,18]. However, for some T2DM patients, the effects of SGLT2i on glycemic control or body weight loss seem to be limited during treatment with this class of antihyperglycemic agents [19]. Brown et al. have brought into discussion compensatory hyperphagia, alterations in energy expenditure and/or preferred substrate as potential limiting mechanisms for weight loss with SGLT2i. This could be in part due to metabolic and hormonal adaptations translating into an increase in hepatic glucose output and fatty acid utilization [20]. Previous randomized controlled trials suggest that poor glycemic control at baseline was related to a good response to SGLT-2i [21,22], while renal dysfunction was associated with reduced efficacy of this medication in improving glycemic control [23].

Recently, a retrospective study on Korean patients reported that a higher baseline of glycated hemoglobin (HbA1c) levels and preserved estimated glomerular filtration rate (eGFR) were predictive markers of better glycemic responses to dapagliflozin [24]; specifically, female and obese T2DM patients tended to lose more weight during the treatment with this SGLT-2i [15]. Another study revealed that the glucose-lowering effect of empagliflozin was more evident in T2DM patients with higher baseline HbA1c levels, better renal function, and shorter T2DM duration. In addition, a higher body mass index (BMI) and lower baseline HbA1c levels were predictive clinical parameters of body weight reduction in patients treated with empagliflozin. In this study, patients with a baseline HbA1c < 7.5% lost an average of 2.4 kg, in contrast with the −2.1 kg reduction obtained by those with an HbA1c between 7 and 9%, with insulin resistance being cited as a possible explanation for this finding [11].

An additional effect that has been reported in some studies with SGLT2 inhibitors associated with weight loss was the lowering of blood pressure, being beneficial for groups of patients with obesity, hypertension or other cardiovascular diseases [25,26,27].

Because the mechanism of action of SGLT2 inhibitors is independent of insulin secretion or action, this class is considered to be complementary to existing drugs for the treatment of diabetes, effective in all stages of the disease, used alone or in combination with metformin, sulphonylureas, thiazolidinediones or insulin in patients with an acceptable renal function [28,29,30]. Patients at high risk of hypoglycemia may benefit from a combination of metformin-SGLT2 inhibitors, as the risk of hypoglycemia of the latter is low compared to treatment with insulin and sulfonylurea derivatives [30].

New directions of use for SGLT2i are the treatment of breast and prostate cancer; studies have been conducted in human breast cancer MCF-7 cells. The SGLT2 inhibitor ipragliflozin reduced breast cancer cell proliferation via membrane hyperpolarization and mitochondrial membrane instability [31].

Alongside their benefits, SGLT2i have shown adverse effects such as diabetic ketoacidosis, genital and urinary infections, volume depletion, increased risk of fracture or amputation, observed for dapagliflozin, empagliflozin, ertugliflozin, and canagliflozin [32,33,34,35,36].

Despite the growing acceptance over the past several years to therapeutically treat T2DM using SGLT2i, there is still a need to better understand the safety and effectiveness of these drugs in this patient population, especially as part of a larger combination therapy regimen [3]. Herein, we performed this clinical investigation to assess the long-term effectiveness and safety of SGLT-2 inhibitors (SGLT-2i) in a Romanian population of subjects affected by T2DM.

We focused on investigating the long-term effectiveness and safety of two distinct SGLT-2i, empagliflozin and dapagliflozin, in inadequately controlled T2DM in combination with metformin, other oral antidiabetic agents or insulin, according to the prescription protocols approved in Romania during the studied period.

## 2. Materials and Methods

### 2.1. Protocol of the Research

The data of a total of 100 patients treated for T2DM were collected and analyzed; the patients were treated at the outpatient Diabetes, Nutrition and Metabolic Diseases Clinic of the ‘Sf. Spiridon’ University Emergency Clinical Hospital and the Consultmed Clinic in Iasi, between 2017 and 2021, with one of the SGLT-2 inhibitors currently available in Romania (dapagliflozin, empagliflozin), either in combination with other oral antidiabetic drugs or insulin (Figure 1).

Eligible patients were over 18 years with T2DM and associating dyslipidemia and/or cardiovascular disease who had been treated with dapagliflozin or empagliflozin, a criterion that divided the study group into two subgroups: Dapa group—48 patients treated with dapagliflozin combined with oral antidiabetics or insulin; Empa group—52 patients treated with empagliflozin combined with oral antidiabetics.

The variables analyzed to assess therapeutical efficacy and safety included: age, gender, time (in years) from the onset of diabetes, duration of treatment (in months), reasons for interrupting treatment, body weight, BMI, blood pressure (BP), blood sugar, HbA1c, serum creatinine, estimated glomerular filtration rate (eGFR) (a dynamic at the beginning as well as at the interruption of treatment or on the last recorded checkup), total serum cholesterol, triglyceride, low-density lipoprotein (LDL-cholesterol), high-density lipoprotein (HDL-cholesterol), alanine aminotransferase (ALT), uric acid.

### 2.2. Research Ethics

The data collection was approved by the board of ‘Sf. Spiridon’ University Emergency Clinical Hospital, Consultmed Clinic Iasi and by the Committee for Research and Ethical Issues of the ‘Grigore T. Popa’ University of Medicine and Pharmacy Iasi (Certificate no. 30/14.01.2021). As the data were analyzed anonymously, written informed consent was not necessary. The study was performed in accordance with the Declaration of Helsinki (https://www.wma.net/policies-post/wma-declaration-of-helsinki-ethical-principles-for-medical-research-involving-human-subjects/ accessed on 11 March 2022).

### 2.3. Statistical Analysis

The data were collected and stored in an SPSS 18.0 software database and processed with the correct statistical functions. In presenting the data, 95% relevance trust gaps were used. For the descriptive statistical analysis, we used the ANOVA test, which showed the following indicators: primary indicators: minimum, maximum, median; median value indicators: simple arithmetical average, median; dispersion indicators: standard deviation (SD), standard error, confidence interval (IC%); skewness test (−2 < *p* < 2)—validates the normality of the series of levels, used when the examined variable shows continuous levels.

Significant quality tests used were: test χ^2^ nonparametric test, which compares two or more frequency repartitions—applied when the events expected are excluded; Kruskal–Wallis nonparametric test, which compares inter-groups of tree or more frequency repartitions.

The study of correlations between various phenomena was carried out with the help of the Pearson correlation quotient, which shows the intensity of statistical connections and their significance. The levels of the correlation quotient are between (−1, +1): if it leans toward +1, there is a great linear dependence of phenomena (direct correlation); if it leans toward −1, there is also a great linear reliance (indirect correlation); the closer it becomes to 0, the smaller the intensity of the connection.

Quantitative relevance tests: t-Student test, which takes into account the measuring of variability and importance of observations. Based on the average and standard deviation for each group studied, we calculate a t. Depending on the number of degrees of freedom (df), if the calculated levels are higher than the charted one, then the zero hypothesis is rejected, and the difference is declared to be “statistically relevant”. The value of *p* is smaller when the relevance is stronger. The F test (the difference between the average and standard intragroup deviation) is an extension of the t-Student test applied to averages of two or more groups. After the ANOVA test, we applied a Bonferroni correction (post hoc Bonferroni). This correction reduces the error ratio when testing several hypotheses.

## 3. Results and Discussion

### 3.1. Evaluation Phase

The study group was balanced, with males accounting for 58.3% of the Dapa patients and 61.5% of the Empa patients, but the differences in percentage were not statistically significant (*p* = 0.744) (Figure 2).

The age levels were homogenous, which suggests that tests for statistic relevance can be applied: variations in the 45–85-years-old range; average 62.04 years ± 7.75; median of 63 years old; skewness test result *p* = −0.068. The average age in the Dapa group was slightly higher compared to those of the Empa group (62.94 vs. 61.21 years; *p* = 0.268). In the intragroup, in male patients treated with Dapa, we noted a significantly lower age average compared to the female group (61.11 vs. 65.50 years; *p* = 0.048), which is also noticed in the Empa group (59.88 vs. 63.35 years; *p* = 0.05).

The distribution of age groups showed that the most Dapa patients were in the 60–69 years age group (45.8%), and over 20% of the group were over 70. A total of 48.1% of the patients from the Empa group were in the 60–69 years group, and 13.5% in the 70–79 years. Adding up the results, we obtained 65 as the age threshold for the subsequent statistical analysis: 39.6% Dapa patients and 42.3% Empa are above this average (*p* = 0.782).

The set of figures for the duration of diabetes was homogenous, which suggests that tests can be applied for statistic relevance: variations between the 2–21 years; group average 8.46 years ±4.74; median of 8 years; skewness test results *p* = 0.686. The average duration of diabetes in the Dapa group was significantly higher than in the Empa group (9.69 vs. 8.46 years; *p* = 0.012). For the intragroup, in the male Dapa patients, a slightly lower average in the duration of diabetes was noticed compared to female patients (9.14 vs. 10.45 years; *p* = 0.350), which is not noticed in Empa patients (7.72 vs. 6.70 years; *p* = 0.433), but the differences are not statistically significant.

The Dapa patients had a higher age average and duration of diabetes slightly higher than the Empa patients. However, in both groups, male patients had a lower age average and shorter duration of diabetes than female patients.

Correlating the patient’s age with the duration of diabetes, we detected the following differences: in the Dapa group, the parameters were apparently independent (r = +0.040; *p* = 0.786); in the Empa group, the correlation was direct and of reduced intensity (r = +0.192; *p* = 0.173), but the result cannot be extended to the general population.

In the cases studied, 93.7% of the Dapa patients and 76.9% of the Empa patients had a duration of diabetes longer than 4 years (*p* = 0.002), but it is important to note that the frequency of cases with a duration of diabetes of 4–10 years was comparable between the two groups (52.1% vs. 57.7%) (Figure 3).

The series of figures for the number of checkups were homogenous, which suggests that tests for statistical relevance can be applied. In the studied groups, we had between 1 and 17 checkups, with an average level of 6.06 ± 3.40 visits, close to the median level for the group (6 visits), significantly higher in the male group (6.83 vs. 4.90; *p* = 0.005). The average number of checkups was not significantly different based on the type of treatment, approximately six in both groups: 6.42 ± 4.11 visits Dapa group and 5.73 ± 2.56 visits Empa group (*p* = 0.315). The number of months of treatment varied between 2 and 50, the average being approximately 18 months ± 310; in the female group, an average of 14.65 months of treatment, in the male group, an average of 20.45 months, which is significantly higher (*p* = 0.004).

The average number of months of treatment with Dapa + metformin was significantly higher compared to the Dapa + insulin treatment (22.36 ± 12.28 vs. 14.70 ± 11.06 months; *p* = 0.036). The average number of months of Empa treatment 10 mg + metformin was not significantly different from the Empa treatment 25 mg + metformin (16.90 ± 8.39 vs. 17.57 ± 6.57 months; *p* = 0.760).

For the Dapa group and the oral combination, the average length of treatment is 22.36 months, without significant differences between genders (23.16 months/8 visits for males and 20.67 months/7 visits for females; *p* = 0.625). For the Dapa and insulin group, the average length of treatment is 14.7 months, significantly more in the male group (21.11 months/7 visits for males and 9.45 months/3 visits for females; *p* = 0.014. For the Empa groups, for the combination of oral antidiabetics, the average treatment is 18.66 months/6 visits in men and 14.80 months/5 visits in women (*p* = 0.076).

Furthermore, 16 interruptions of treatment were recorded, 9 in women (5 in Dapa group and 5 in Empa group) all due to urinary and genital infections and 7 in men, 4 because of urinary tract infections (1 in Dapa group and 3 in Empa group) and 3 for different reasons (lack of efficiency on glycemic control or SARS-CoV-2 infection, without significant statistical differences between the genders (*p* = 0.152)).

It is noted that in the group of Dapa patients, the duration of treatment was shorter in the elderly patients compared to the Empa patients (Table 1).

Referring to the duration of time spent on treatment, this was higher for the Dapa group compared with the Empa group. However, a comment is required: men had a longer time spent on treatment and fewer adverse reactions causing treatment cessation. Of the 16 treatment discontinuations, all 9 reported in women were due to urinary or genital tract infections, while all 7 registered in the men were due to other reasons (lack of glycemic efficacy and a SARS-CoV-2 infection), and only 4 were due to urinary infections. In terms of the geriatric population, time on treatment was longer for Empa compared to Dapa.

The levels for the BMI were homogenous, which suggests that tests of statistical relevance can be applied: variations between 24.75 and 48.10 kg/m^2^; group average 33.15 kg/m^2^ ± 4.90; median of 32.69 kg/m^2^; skewness test result *p* = 1.013. The average BMI for the Empa group was significantly higher compared to the one of the Dapa group (34.16 vs. 32.04 kg/m^2^; *p* = 0.03). For the intragroup, among male patients on Dapa, a slightly lower average level of BMI was recorded compared to the female group (31.61 vs. 32.65 kg/m^2^; *p* = 0.434), a difference that, in the Empa group, was statistically significant (32.83 vs. 36.30 kg/m^2^; *p* = 0.015).

The average BMI decreased significantly after Dapa treatment (32.04 vs. 31.40 kg/m^2^; *p* = 0.006), especially in the male group (31.61 vs. 30.92 kg/m^2^; *p* = 0.023) and in the group treated on Dapa+metformin (32.54 vs. 31.93 kg/m^2^; *p* = 0.045) or Dapa+insulin (31.34 vs. 30.67 kg/m^2^; *p* = 0.001) (Table 2).

In the Dapa group, body weight decreased by 2.1%, with treatment associated with oral antidiabetics as well as in association with insulin, while BMI decreased by 1.8% in the Dapa group and 2.1% in the Empa group. The average BMI lowered significantly after Empa treatment (34.31 vs. 33.35 kg/m^2^; *p* = 0.003), in male groups (32.83 vs. 31.81 kg/m^2^; *p* = 0.018), as well as female groups (36.30 vs. 35.36 kg; *p* = 0.042) and those on Empa 10 mg+metformin (34.64 vs. 33.14 kg/m^2^; *p* = 0.002) or Empa 25 mg+metformin (33.46 vs. 33.22 kg/m^2^; *p* = 0.002) (Table 2).

Patients’ body weight decreased significantly after Dapa treatment (88.28 vs. 86.42 kg; *p* = 0.006) in males (94.13 vs. 91.96 kg; *p* = 0.02) as well in females (80.10 vs. 78.65 kg; *p* = 0.05) and in the group treated with Dapa + metformin (90.04 vs. 88.11 kg; *p* = 0.04) (Table 2).

In the Empa+metformin group, body weight diminished by approximately 4.4%, while BMI decreased by approximately 4.3%. In the Empa 25 mg+metformin group, body weight decreased only by 0.8%, and BMI decreased by 0.7%, without being justified by the data in the observation sheets (Table 2).

### 3.2. Reevaluation Phase

The average BMI in the Empa group of patients was higher than in the Dapa group, and with both, it was lower in men than in women at the beginning of the study as well as at reevaluation (*p* < 0.05). However, after treatment, the decrease in BMI was higher in men vs. women in the Dapa group as well as the Empa group, with a more pronounced reduction overall in the Empa group.

Depending on the treatment received, upon reevaluation, the changes were (Figure 4: in patients treated with Dapa+metformin, the reduction in BMI was insignificant (32.54 ± 4.93 vs. 31.93 ± 4.67kg/m^2^; *p* = 0.045); in patients on Dapa+insulin, the decrease in BMI was insignificant (31.34 ± 3.81 vs. 30.67 ± 3.59 kg/m^2^; *p* = 0.073); overall, in the group treated with Dapa, the diminution of BMI was highly significant (32.04 ± 4.49 vs. 31.40 ± 4.18 kg/m^2^; *p* = 0.006); in patients treated on Empa 10 mg+metformin, the reduction in BMI was significant (34.64 ± 5.80 vs. 33.14 ± 5.70 kg/m^2^; *p* = 0.002); in patients treated with Empa 25 mg+metformin, the decrease in BMI was insignificant (33.45 ± 3.82 vs. 33.22 ± 3.84 kg/m^2^; *p* = 0.421); overall in the group treated on Empa, the decrease in BMI was significant (34.16 ± 5.08 vs. 33.17 ± 4.99 kg/m^2^; *p* = 0.002) (Figure 4).

In patients treated with Dapa, the evolution of BMI varied between −5.05 and +1.07 kg/m^2^, while in patients treated with Empa, it was between −8.55 and +1.77 kg/m^2^, with significant differences in average levels: Dapa group −0.63 ± 1.54 kg/m^2^, Empa group −0.99 ± 2.16 kg/m^2^ (*p* = 0.004).

Patients’ weight decreased significantly in patients of both genders after Empa treatment but especially after Dapa treatment.

At the beginning of the study, 81.3% of patients in the Dapa group and 82.7% of those treated on Empa (*p* = 0.851) were diagnosed, according to the observation sheets, with various stages of hypertension (HT).

In our study, we noticed a significant lowering of systolic BP values in both groups, more significant in women (Dapa group: 143 vs. 136 mmHg; *p* = 0.001 and Empa group: 139 vs. 133 mmHg; *p* = 0.05). Furthermore, we found important systolic BP reductions in the following subgroups: Dapa+oral antidiabetes agents and Empa+oral antidiabetes agents. The effect of SGLT2 inhibition on BP reduction translates into 3–7 mmHg for systolic and 2 mmHg for diastolic values, as mentioned in the literature. The mechanisms behind this effect relate to osmotic diuresis, body weight reduction and changes in the sympathetic nervous system [37,38].

In patients from the Dapa group, after comparing the figures of BP at the moment of inclusion in the study with the staging in the observation sheet, the following variations were noted (*p* = 0.110) (Figure 5): 22.2% of subjects had normal blood pressure, 33.3% were diagnosed with stage I HT, 14.3% with stage II HT, respectively, and 5% of them with stage III HT. In patients treated with Empa, comparing the BP at the moment of joining the study with the staging in the medical sheet, the following variations were shown (*p* = 0.379) (Figure 5): in 11.1% of subjects, the modifications in blood pressure values were not observed, 44.4% were identified with stage I HT, 22.7% with stage II HT, and the rest of them with stage III HT. Though they had different stages of HT, the patients in both groups received different antihypertensive drugs.

Comparing the groups under observation according to HT staging, there were no statistical differences in the distribution of cases based on gender, or age (*p* > 0.05) (Table 3).

Systolic BP decreased significantly after Dapa treatment (141 vs. 133 mmHg; *p* = 0.001), especially in the group treated on Dapa+metformin (139 vs. 131 mmHg; *p* = 0.003). Systolic BP decreased significantly after Empa treatment (140 vs. 133 mmHg; *p* = 0.021), especially in the female group (139 vs. 133 mmHg; *p* = 0.05) and the Empa 10 mg+metformin (140 vs. 131 mmHg; *p* = 0.029).

The evolution of average glomerular filtration rate (GFR) levels after Dapa treatment was not significant, in men (99.33 vs. 99.25 mL/min; *p* = 0.981), or women (91.25 vs. 84.20 mL/min; *p* = 0.191). Before treatment, the average of GFR levels was not significantly different between genders (99.33 vs. 91.25 mL/min; *p* = 0.221), and after the Dapa treatment, the GFR average levels were significantly higher in the male group (99.25 vs. 84.20 mL/min; *p* = 0.006) (Table 4).

In the Empa group, before treatment, the average GFR levels were significantly different between the sexes (99.77 vs. 87.95 mL/min; *p* = 0.014). The evolution of average GFR levels after Empa treatment was significantly increased in men (99.77 vs. 106.0 mL/min; *p* = 0.023), and in women (87.95 vs. 96.05 mL/min; *p* = 0.047) (Table 4).

Depending on the treatment, on reevaluation. the following changes in GFR were noted (Figure 6): in patients receiving Dapa+metformin, the decrease in GFR was insignificant (103.94 ± 23.45 vs. 99.54 ± 19.95 mL/min; *p* = 0.225); in patients receiving Dapa+insulin, the reduction in GFR was insignificant (85.50 ± 16.05 vs. 85.03 ± 14.08 mL/min; *p* = 0.878); overall, in those treated with Dapa, the decrease in GFR was insignificant (95.06 ± 22.36 vs. 93.49 ± 19.0 mL/min; *p* = 0.293); in patients receiving Empa 10 mg+metformin, the increase in GFR was significant (95.99 ± 15.03 vs. 102.87 ± 19.64 mL/min; *p* = 0.010); in patients receiving Empa 25 mg+metformin, the increase in GFR was insignificant (94.10 ± 20.14 vs. 101.0 ± 24.23 mL/min; *p* = 0.087); for the overall group treated with Empa, the intensification of GFR was significant (95.22 ± 17.11 vs. 102.12 ± 21.40 mL/min; *p* = 0.002).

In patients treated with Dapa, the evolution of GFR varied between −47 and +31 mL/min, while in patients treated with Empagliflozin, the variation was between −24 and +61 mL/min, with important differences in averages: Dapa group −2.46 ± 16.11 mL/min, Empa group 6.88 ± 15.39 mL/min (*p* = 0.004). However, in the first 2–3 months of treatment, slight differences in GFR were shown, after which this difference increases in the first 2 years.

The frequency of cases with a decrease in GFR in the first year is 27.3% in the Dapa group and 30% in the Empa group.

Before SGLT2i treatment, eGFR baseline levels did not differ significantly between men and women for the Dapa group. However, this was not the case for the Empa group, in which initial eGFR differences were important. After being treated with SGLT2i, the only significant eGFR increase was noticed in the Empa group (95.22 ± 17.11 vs. 102.12 ± 21.40 mL/min; *p* = 0.002). Correlating the dynamic modifications of the eGFR with the number of months of treatment, the evolution of these parameters was apparently independent.

Regardless of the treatment, the average blood glucose levels were higher in men before as well as after treatment.

In the Dapa group, the correlations between the duration of the diabetes and GFR levels were indirect, moderate in intensity, and statistically relevant (r= −0.326; *p* = 0.024), but in the Empa group, the two parameters seem to be independent (r = −0.055; *p* = 0.696).

Regarding the blood sugar levels and the HbA1c values, they decrease significantly after the treatment in both male and female groups, with a strong reduction of HbA1c in male patients.

Before the Dapa treatment, blood sugar average levels were significantly higher in the male group (170 vs. 148 mg/dL; *p* = 0.043), but after Dapa treatment, they decreased significantly (170 vs. 136 mg/dL; *p* = 0.001), reaching the average levels of the female group (136 vs. 145 mg/dL; *p* = 0.280) (Table 4).

After Empa treatment, average levels of blood sugar decreased significantly (163 vs. 140 mg/dL; *p* = 0.002), mostly in the female groups (171 vs. 139 mg/dL; *p* = 0.001) (Table 4).

After Empa treatment, average levels of HbA1c diminished significantly per total group studied (7.72% vs. 7.35%; *p* = 0.004), in the male groups (7.71% vs. 7.30%; *p* = 0.029), and female groups (7.75% vs. 7.45%; *p* = 0.039) (Table 4).

Coronary syndrome was slightly more frequent in patients in the Dapa group (45.8% vs. 38.5%; *p* = 0.455) (Figure 7).

Heart failure was diagnosed in 20.8% of the patients treated with Dapa and 17.3% of the patients treated with Empa (*p* = 0.654) (Figure 7).

To study the link between prediction and response, we have constructed a mathematical model to determine whether glucose, eGFR, and HbA1c may be good predictors of determinism of heart failure after treatment with Dapa or Empa. The ROC (Receiver Operating Characteristics) curve is a two-dimensional curve in which we have sensitivity on the Y-axis and specificity on the X-axis. This curve helps us measure the efficiency of the model. The larger the area under the curve (the maximum value is 1), the better the model.

When highlighting the receiver operating characteristic curve (ROC) curve, we noted that HbA1c stands out as a good predictor of heart failure in patients treated with Dapa (area under the curve AUC = 0.617: IC95%: 0.446–0.788; *p* = 0.259), and GFR as a good predictor of heart failure in Empa patients (AUC = 0.647: IC95%: 0.461–0.843; *p* = 0.168) (Figure 8).

In the cases studied, the ROC curve only confirms BMI as a good predictor in selecting SGLT2 treatment, with threshold levels of 34.1 with 65% sensitivity and 56% specificity (AUC = 0.632; IC95%) (Figure 9).

SGLT2i have been studied in various clinical trials for their benefit in preventing heart failure [39,40,41]. Recent research has shown a cardio-renal mechanism of inhibition of Na^+^/H^+^ exchanger (NHE) isoforms: NHE1 isoform in the myocardium, with a further effect of reducing cytoplasmic ions calcium and sodium levels and NHE3 in the proximal renal tube, which mediates tubular sodium reuptake [42,43]. Another mechanism by which sodium ion is reduced is the inhibition of adipokine leptin by SGLT2i (effect demonstrated for canagliflozin), combined with the pro-activation of anti-inflammatory adipokines (e.g., adiponectin), thus preventing heart failure [44]. Furthermore, studies on rats have demonstrated a positive effect of some SGLT2i in preventing and even reducing cardiac fibrosis (e.g., dapagliflozin, empagliflozin) by interfering with the synthesis of collagen; more interestingly, it appears that this effect is independent of hyperglycemia [45,46].

Their cardiovascular and renal protection effects in diabetic and non-diabetic patients are good premises for the SGLT2i class to be put in the category of drugs for cardiorenal protection in the future [47,48,49,50,51].

## 4. Conclusions

Taking into account the results obtained, we can say that a better result in weight loss, decrease in BMI, reduction in BP and HbA1c, and increase in average GFR values could be obtained in younger male patients with a shorter duration of diabetes and threshold BMI levels of 34.1, treated with SGLT2, more significantly with Empa.

Of course, considering the limitations of prescription protocols in Romania in the period of study, which did not allow the combination of Empa with insulin, as well as the size of the group studied, these results cannot be extended to the general population. Nevertheless, the study confirms, from certain points of view, the communications of other recent studies in the field and can be used as a starting point in future analyses for individualizing the approach in the treatment of T2DM.

## Figures and Tables

**Figure 1 healthcare-10-01153-f001:**
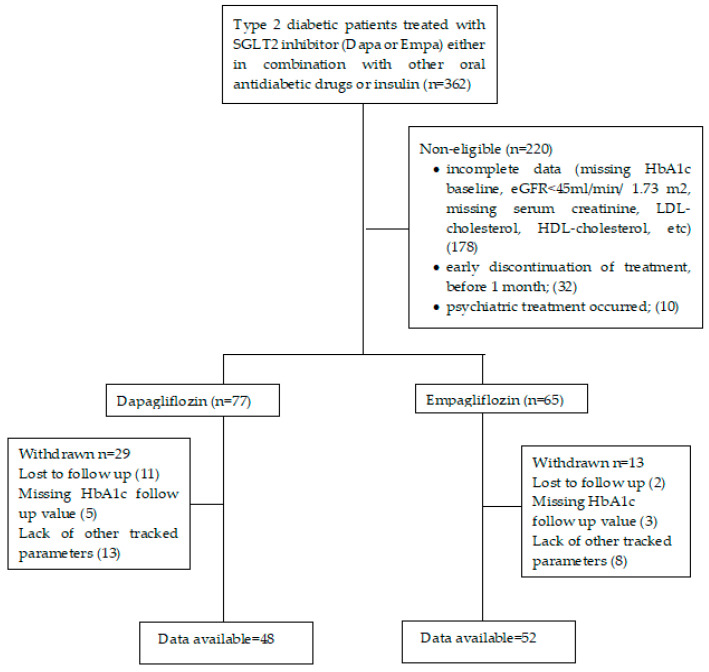
Flow chart of the study.

**Figure 2 healthcare-10-01153-f002:**
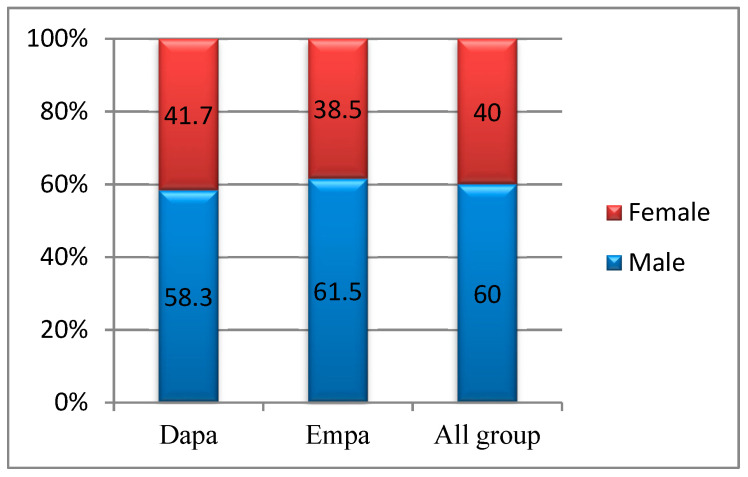
Group distribution by gender.

**Figure 3 healthcare-10-01153-f003:**
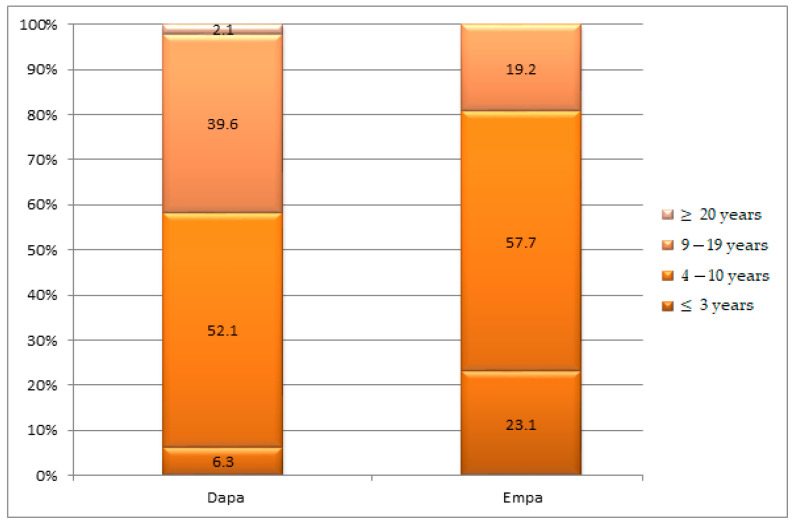
Group structures according to the age of diabetes in years.

**Figure 4 healthcare-10-01153-f004:**
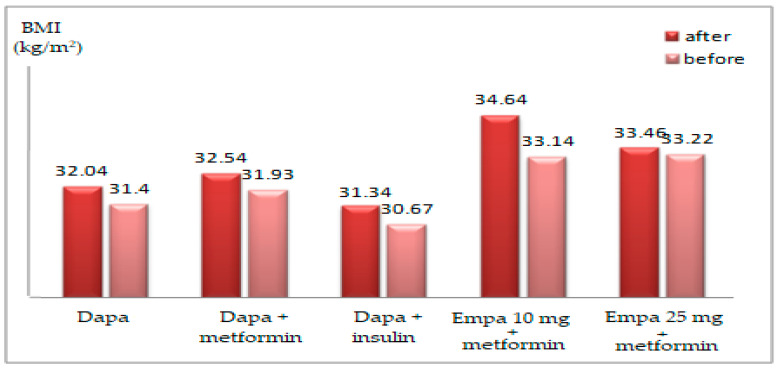
The mean BMI evolution according to the treatment administered.

**Figure 5 healthcare-10-01153-f005:**
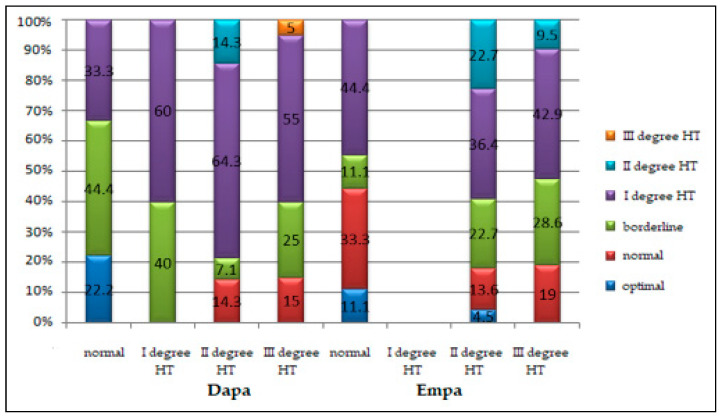
Correlation of BP values measured at the study entry with comparative staging by study groups.

**Figure 6 healthcare-10-01153-f006:**
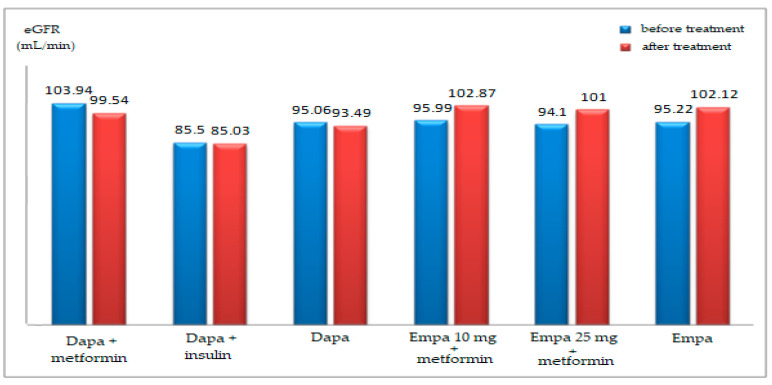
Average values evolution of the eGFR depending on the treatment administered.

**Figure 7 healthcare-10-01153-f007:**
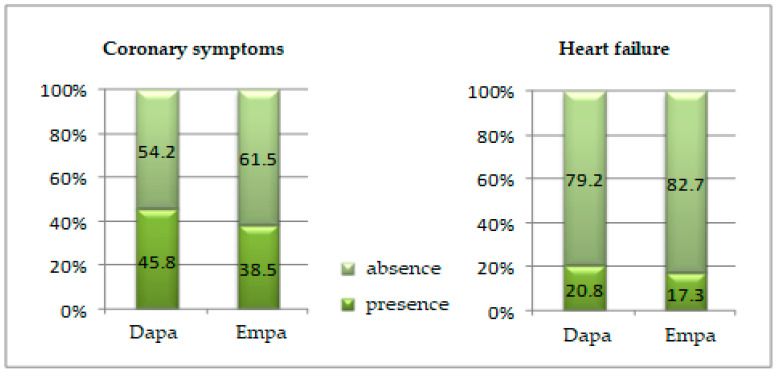
Comparison of the coronary syndrome and heart failure presence in the groups studied.

**Figure 8 healthcare-10-01153-f008:**
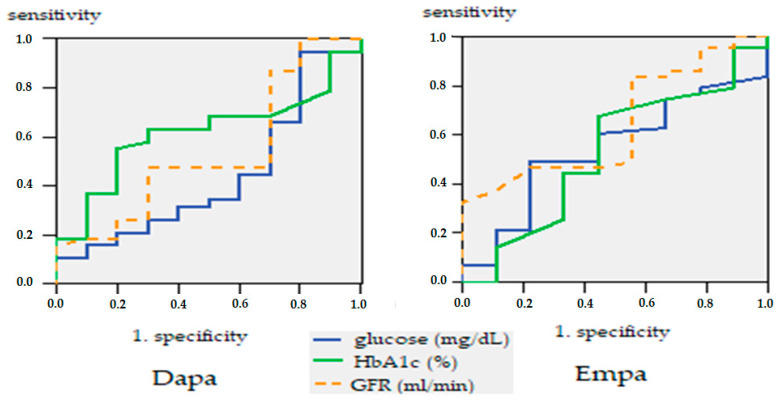
ROC curve, glucose, HbA1c and GFR are predictors of heart failure.

**Figure 9 healthcare-10-01153-f009:**
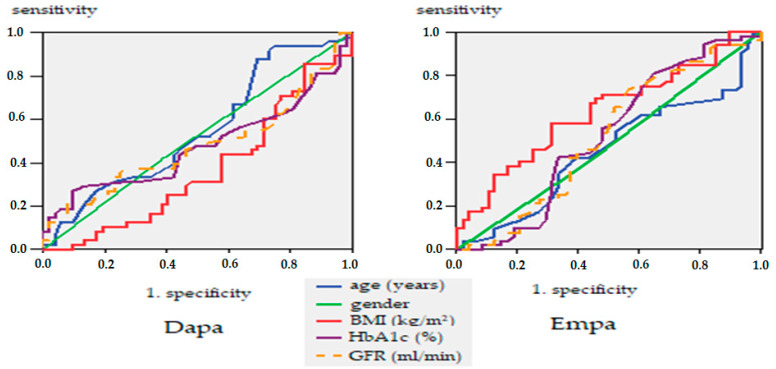
ROC curve. Prognostic factors in the selection of SGLT-2 treatment.

**Table 1 healthcare-10-01153-t001:** Treatment duration until discontinuation.

Age Groups	Period of Treatment		
	2–3 Months	6–12 Months	1–3 Years
Dapa group			
40–49 years			
50–59 years		2	
60–69 years		2	1
over 70 years	2		1
Empa group			
40–49 years	2		
50–59 years		1	1
60–69 years		1	2
over 70 years			1

**Table 2 healthcare-10-01153-t002:** The evolution of parameters after Dapa and Empa treatment, respectively.

Parameters	Before Dapa Treatment			After Dapa Treatment		
	Males	Females	All Cases	Males	Females	All Cases
Weight (kg)						
mean ± SD	94.13 ± 12.23	80.10 ± 13.78	88.28 ± 14.54	91.96 ± 10.74 ^(0.02)^	78.65 ± 12.41 ^(0.05)^	86.42 ± 13.13 ^(0.006)^
t-Student (*p*)	**0.001**		**0.001**			
BMI (kg/m^2^)						
mean ± SD	31.54 ± 3.63	32.31 ± 4.09	31.86 ± 3.81	31.21 ± 3.78 ^(ns)^	32.68 ± 2.77 ^(ns)^	31.82 ± 3.45 ^(ns)^
t-Student (*p*)	0.493		0.159			
systolic BP (mmHg)						
mean ± SD	140 ± 18	143 ± 8	141 ± 15	131 ± 12 ^(ns)^	136 ± 15 ^(0.001)^	133 ± 13 ^(0.001)^
t-Student (*p*)	0.466		0.243			
**Parameters**	**Before Empa treatment**			**After Empa treatment**		
	**Males**	**Females**	**All cases**	**Males**	**Females**	**All cases**
Weight (kg)						
mean ± SD	97.16 ± 13.98	90.30 ± 21.80	94.52 ± 17.53	94.03 ± 13.27 ^(0.019)^	87.80 ± 20.47 ^(0.034)^	91.63 ± 16.51 ^(0.002)^
t-Student (*p*)	**0.047**		**0.048**			
BMI (kg/m^2^)						
mean ± SD	33.24 ± 4.19	36.01 ± 6.09	34.31 ± 5.13	32.07 ± 4.13 ^(0.016)^	35.89 ± 5.80 ^(ns)^	33.35 ± 5.06 ^(0.003)^
t-Student (*p*)	0.058		**0.020**			
systolic BP (mmHg)						
mean ± SD	140 ± 15	139 ± 11	140 ± 13	136 ± 18 ^(ns)^	133 ± 14 ^(0.05)^	133 ± 13 ^(0.021)^
t-Student (*p*)	0.733		0.583			

ns: non-significant; *p* < 0.05; *p* < 0.01.

**Table 3 healthcare-10-01153-t003:** Correlation of demographic characteristics with HT compared to groups studied.

Characteristics	Stage I HT	Stage II HT	Stage III HT	*p* *
Males				
Dapa group	1 (100%)	8 (34.8%)	13 (54.2%)	0.327
Empa group	0 (0.0%)	15 (65.2%)	11 (45.8%)	
Females				
Dapa group	4 (100%)	6 (46.2%)	7 (41.2%)	0.104
Empa group	0 (0.0%)	7 (53.8%)	10 (58.8%)	
under 65 years				
Dapa group	3 (100%)	8 (40.0%)	12 (52.2%)	0.263
Empa group	0 (0.0%)	12 (60.0%)	11 (47.8%)	
over 65 years				
Dapa group	2 (100%)	6 (37.5%)	8 (44.4%)	0.167
Empa group	0 (0.0%)	10 (62.5%)	10 (55.5%)	

*, Kruskal–Wallis test.

**Table 4 healthcare-10-01153-t004:** Evolution of laboratory parameters after Dapa and Empa treatment, respectively.

Parameters	Before Dapa Treatment			After Dapa Treatment		
	Males	Females	All Cases	Males	Females	All Cases
blood glucose (mg/dL)						
mean ± SD	170 ± 39	148 ± 32	161 ± 37	136 ± 29 ^(0.001)^	145 ± 28 ^(ns)^	140 ± 29 ^(0.001)^
t-Student (*p*)	**0.043**			0.280		
HbA1c (%)						
mean ± SD	7.83 ± 1.16	7.99 ± 1.48	7.90 ± 1.29	7.41 ± 0.86 ^(0.010)^	7.65 ± 1.19 ^(ns)^	7.51 ± 1.00 ^(0.010)^
t-Student (*p*)	0.685			0.439		
GFR (mL/min)						
mean ± SD	99.33 ± 16.80	91.25 ± 28.21	95.96 ± 22.36	99.25 ± 20.56 ^(ns)^	84.20 ± 13.47 ^(ns)^	92.98 ± 19.30 ^(ns)^
t-Student (*p*)	0.221			**0.006**		
**Parameters**	**Before Empa treatment**			**After Empa treatment**		
	**Males**	**Females**	**All cases**	**Males**	**Females**	**All cases**
blood glucose (mg/dL)						
mean ± SD	158 ± 35	171 ± 37	163 ± 36	145 ± 33 ^(ns)^	139 ± 22 ^(0.001)^	140 ± 29 ^(0.002)^
t-Student (*p*)	0.216			0.485		
HbA1c (%)						
mean ± SD	7.71 ± 0.82	7.75 ± 0.53	7.72 ± 0.72	7.30 ± 0.74 ^(0.029)^	7.45 ± 0.65 ^(0.039)^	7.35 ± 0.71 ^(0.004)^
t-Student (*p*)	0.863			0.467		
GFR (mL/min)						
mean ± SD	99.77 ± 16.96	87.95 ± 15.05	95.22 ± 17.11	106 ± 20.8 ^(0.023)^	96.05 ± 21.44 ^(0.047)^	102 ± 21.4 ^(0.002)^
t-Student (*p*)	**0.014**			0.107		

ns: non-significant; *p* < 0.05; *p* < 0.01.

## Data Availability

Not applicable.

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
