# Peer review of "Clinical Parameters Affecting the Therapeutic Efficacy of SGLT-2—Comparative Effectiveness and Safety of Dapagliflozin and Empagliflozin in Patients with Type 2 Diabetes"

_healthcare, 2022, doi:10.3390/healthcare10071153_

Round 1

Reviewer 1 Report

The manuscript entitled Clinical Parameters Affecting the Therapeutic Efficacity of 2 SGLT-2 -Comparative Effectiveness and Safety of 3 Dapagliflozin and Empagliflozin in Patients with Type 2  Diabetes by Irina Claudia Anton et al. showed that the reduction of HbA1c with SGLT2 inhibitors (two SGLT-2 inhibitors: empagliflozin  and dapagliflozin ), combined with metformin, other oral antidiabetics or insulin, was greater in patients with a lower mean age.The manuscript is well written, and the results presented may be used to improve blood glucose levels in the patient with type 2 diabates. Therefore, I consider this manuscript should be published after major revision. However, below are some suggestions that which authors should address?

  1. There is a redundancy in the references in the introduction section. For example, references 11 and 12 are after reference 7.
  2. The introduction section should be elaborated as there is enough literature about the role of SGLT2 inhibitors in type 2 diabetes. Such as https://www.ncbi.nlm.nih.gov/pmc/articles/PMC6028052/. So introduction should be added with more recent references.
  3. In the result section figures 3 and 4 should be combined so that results will be clear to the readers. Similar Figures 8 and 9 should also be combined.
  4. Figure qualities should be improved, and results are present in parts they should combine and results should be precise.
  5. The discussion part is poorly written, as this is not supported by the references. And my suggestion is dissuasion part should be combined with the result section
  6. The effect on cardiovascular diseases and blood pressure by  SGLT2 inhibitors should be discussed in the discussion section.

Author Response

Reviewer 1

The manuscript entitled Clinical Parameters Affecting the Therapeutic Efficacy of 2 SGLT-2 -Comparative Effectiveness and Safety of Dapagliflozin and Empagliflozin in Patients with Type 2  Diabetes by Irina Claudia Anton et al. showed that the reduction of HbA1c with SGLT2 inhibitors (two SGLT-2 inhibitors: empagliflozin  and dapagliflozin ), combined with metformin, other oral antidiabetics or insulin, was greater in patients with a lower mean age.The manuscript is well written, and the results presented may be used to improve blood glucose levels in the patient with type 2 diabetes. Therefore, I consider this manuscript should be published after major revision. However, below are some suggestions that which authors should address?

  1. There is a redundancy in the references in the introduction section. For example, references 11 and 12 are after reference 7.

The error has been corrected in accordance with the reviewer’s comment.

  1. The introduction section should be elaborated as there is enough literature about the role of SGLT2 inhibitors in type 2 diabetes. Such as https://www.ncbi.nlm.nih.gov/pmc/articles/PMC6028052/. So introduction should be added with more recent references. Please see the corrected version in the attachment below.

We wish to express our deep appreciation to the reviewer for his remark regarding the other communicated data in the field of our research. We have improved the discussion section by adding important information based on the suggested link and other papers. WE have modified the references accordingly.

  1. In the result section figures 3 and 4 should be combined so that results will be clear to the readers. Similar Figures 8 and 9 should also be combined.

It is not possible to combine these figures, because the conveyed information will be difficult for the reader to understand.

  1. Figure qualities should be improved, and results are present in parts they should combine and results should be precise.

We have amended the accuracy of some figures.

  1. The discussion part is poorly written, as this is not supported by the references. And my suggestion is dissuasion part should be combined with the result section.

In accordance with the reviewer’s comment, we have combined the results and discussions sections and this part has been substantially improved.

  1. The effect on cardiovascular diseases and blood pressure by SGLT2 inhibitors should be discussed in the discussion section.

We thank the reviewer for this suggestion. Some recent information about the effects of SGLT2 inhibitors have been introduced in the manuscript with the corresponding references.

Reviewer 2 Report

Reviewer comments and suggestions

The study evaluated the long-term efficacy and safety in uncontrolled type 2 diabetes (T2DM) by two SGLT-2 inhibitors: empagliflozin (Empa) and dapagliflozin (Dapa), combined with metformin, other oral antidiabetics or insulin. The study included 100 patients retrospectively reviewed (2017-2021) treated for T2DM with associated risk conditions at the University Hospital. 48 patients that received dapagliflozin (10 mg with oral antidiabetics or insulin) and 52 patients, empagliflozin (10 mg /25 mg with oral antidiabetics) were recruited in this study. The result of this study highlighted both groups lowering of BMI was significant, blood sugar average levels decreased significantly (170 vs 136 mg/dL; p = 0.001 for Dapa; 163 vs 140 mg/dL; p = 0,002 for Empa) along with HbA1c (7,90% vs 7,51%; p = 0,01 for Dapa; 7.72% vs 7.35%; p = 0.004 for Empa). The study concluded that better results in all variables were observed in younger male patients with a shorter duration of diabetes.

The paper required a thorough revision to reach the standard of this journal. I have made several suggestions and I advise the authors to modify their manuscript based on the below-provided comments.

  1. In abstract, line 35 is important to highlight here “threshold BMI levels of 43.1, treated”
  2. Line 41 please cite some studies
  3. Line 50-51 more references could be cited here
  4. Line 59 is there was any possible reason for this
  5. Line 70-71 How does the author discusses this study for lower baseline HBA1C level (up to what) as a maker of body weight
  6. Line 2.1 Needs a flow chart to understand the design of this study
  7. Figure 2 figure should be drawn in a professional way and I have noticed that inside the figure text format was not appropriate
  8. I also recommend to the authors to reshape the figure 4 for a better representation. 
  9. First para of the discussion please mention the novelty here in the first para of the discussion
  10. “significantly fewer adverse reactions which would cause an interruption of treatment” could not understand the points
  11. Line 409-410 please elaborate on these lines
  12. Line 416 please explain comprehensively
  13. Line 422 no sufficient discussion concerning ROC curve 
  14. The authors need to add more recent published studies in the discussion section and discuss their findings to validate this present study. 

Author Response

Reviewer 2

The study evaluated the long-term efficacy and safety in uncontrolled type 2 diabetes (T2DM) by two SGLT-2 inhibitors: empagliflozin (Empa) and dapagliflozin (Dapa), combined with metformin, other oral antidiabetics or insulin. The study included 100 patients retrospectively reviewed (2017-2021) treated for T2DM with associated risk conditions at the University Hospital. 48 patients that received dapagliflozin (10 mg with oral antidiabetics or insulin) and 52 patients, empagliflozin (10 mg /25 mg with oral antidiabetics) were recruited in this study. The result of this study highlighted both groups lowering of BMI was significant, blood sugar average levels decreased significantly (170 vs 136 mg/dL; p = 0.001 for Dapa; 163 vs 140 mg/dL; p = 0,002 for Empa) along with HbA1c (7,90% vs 7,51%; p = 0,01 for Dapa; 7.72% vs 7.35%; p = 0.004 for Empa). The study concluded that better results in all variables were observed in younger male patients with a shorter duration of diabetes.

The paper required a thorough revision to reach the standard of this journal. I have made several suggestions and I advise the authors to modify their manuscript based on the below-provided comments.

  1. In abstract, line 35 is important to highlight here “threshold BMI levels of 43.1, treated”

We have made the change in the text.

  1. Line 41 please cite some studies

Some recent studies in the field have been cited at line 41.

  1. Line 50-51 more references could be cited here

We have cited more references in the field.

  1. Line 59 is there was any possible reason for this

Some authors bring into discussion compensatory hyperphagia, alterations in energy expenditure and/or preferred substrate as potential limitative mechanisms for weight loss with SGLT2i. This could be in part due to metabolic and hormonal adaptations translating into an increase in hepatic glucose output and fatty acid utilization.

We have mentioned this explanation in the manuscript. Please see in the attachment below the corrected version.

  1. Line 70-71 How does the author discusses this study for lower baseline HBA1C level (up to what) as a maker of body weight

  1. Line 2.1 Needs a flow chart to understand the design of this study

The flow chart has been included in the manuscript (Figure 1).

  1. Figure 2 figure should be drawn in a professional way and I have noticed that inside the figure text format was not appropriate

We are grateful to the reviewer for this important remark. We have modified the text format inside the figure 2 accordingly.

  1. I also recommend to the authors to reshape the figure 4 for a better representation.

We have reshaped the figure 4.

  1. First para of the discussion please mention the novelty here in the first para of the discussion

We thank the reviewer for his insightful comment on this point. We have reconstructed and improved this part of the manuscript using appropriate references.

  1. “significantly fewer adverse reactions which would cause an interruption of treatment” could not understand the points

We have mentioned this explanation in the manuscript.

  1. Line 409-410 please elaborate on these lines

As for the variation of TAS in the two groups, it had a significantly more important decrease in women and higher in the subgroups on Dapa+oral antidiabetics and Empa (10 mg)+oral antidiabetics, respectively.

We have rephrased (TAS represents the abbreviation for blood pressure in Romanian, accidentally had been left un-translated). The corrected text is:

As for the variation of BP in the two groups, it had a significantly more important decrease in women and higher in the subgroups on Dapa+oral antidiabetics and Empa (10 mg)+oral antidiabetics, respectively.

  1. Line 416 please explain comprehensively

Correlating the dynamic modifications of the eGFR with the number of months of treatment, the evolution of these parameters was apparently independent.

  1. Line 422 no sufficient discussion concerning ROC curve 

To study the link between prediction and response, we have constructed a mathematical model to determine whether glucose, eGFR, and HbA1c may be good predictors of determinism of heart failure after treatment with Dapa or Empa. The ROC (Receiver Operating Characteristics) curve is a two-dimensional curve in which we have sensitivity on the Y axis and specificity on the X axis. This curve helps us measure the efficiency of the model. The larger the area under the curve (the maximum is 1), the better the model.

  1. The authors need to add more recent published studies in the discussion section and discuss their findings to validate this present study. 

We thank the reviewer for his insightful comment on this point. We have reconstructed and improved this part with appropriate references.

Round 2

Reviewer 1 Report

The authors have responded to the reviewers comments and now the manuscript has improved. thus Manuscript should be  be accepted in present form. 

Author Response

Firstly, I would like to express my gratitude to the editor and reviewers for their kind comments, constructive criticisms and valuable suggestions, which helped me to improve my manuscript's overall quality. I have made the changes, as advised, and the revisions are indicated as track changes in the re-revised manuscript. We have removed some figures (number 3, 4, 6, 7, 8, 9, 11, 13, 15, 16, 17), the results being mentioned in the text. All the existing figures in the manuscript have the same histogram.

Reviewer 2 Report

All comments has been incorporated by the authors. Thanks

Author Response

(The authors gave the same response as above.)
